# EveryQuery: Zero-Shot Clinical Prediction via Task-Conditioned Pretraining over Electronic Health Records

**Payal Chandak** [1]  **Gregory Kondas** [2]  **Isaac Kohane** [3]  **Matthew McDermott** [2]

## Abstract

Autoregressive foundation models for electronic health records (EHRs) show strong performance in zero-shot inference across diverse clinical tasks (Waxler et al., 2025; Renc et al., 2024; 2025). These methods generate possible "synthetic futures" for a patient and then infer downstream predictions over the simulated trajectories. Despite their strengths, they suffer from three key flaws: (1) inference is computationally expensive, as each prediction requires simulating many trajectories; (2) performance is noisy due to simulation variance, which especially reduces efficacy on the rare events that are often of high clinical importance; and (3) they are not promptable, as their only input is the patient's medical history, yielding task-agnostic representations. We introduce EveryQuery, a novel EHR foundation model that addresses all three. EveryQuery leverages *task-conditioned pretraining* over random samples of (target outcome, duration) pairs to enable direct, prompted zero-shot inference. Across three datasets, EveryQuery offers consistent improvements in both downstream task AUC (~3–15%) and inference speed (~5,500 to 10,000 times faster) over a competitive autoregressive baseline, while producing a richly structured, task-conditioned embedding space with strong prompt specificity.

## 1. Introduction

Foundation models for structured electronic health record (EHR) data are an emerging paradigm in clinical AI. Like in other domains, they are appealing because they enable *zero-shot inference*: predictions for novel downstream tasks at

[1]Harvard-MIT Program in Health Sciences and Technology [2]Columbia University [3]Harvard Medical School. Correspondence to: Matthew McDermott <mattmcdermott8@gmail.com>.

*Proceedings of the $2^{nd}$ ICML Workshop on Foundation Models for Structured Data*, Seoul, South Korea. 2026. Copyright 2026 by the author(s).

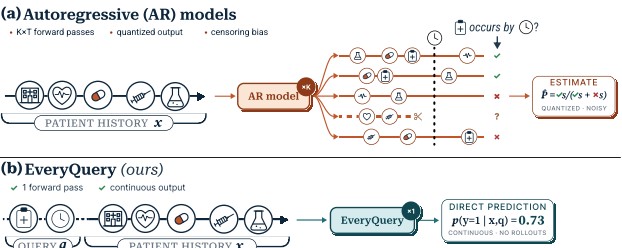

*Figure 1.* **Overview of EveryQuery.** Make a prediction for task query $q = (c, \Delta t)$ on patient history $x$. Autoregressive models **(a)** learn $p(x)$ and achieve zero-shot inference by generating many synthetic futures and aggregating statistics; this yields quantized, high-variance estimates and takes 5,000–10,000× longer. EveryQuery **(b)** learns $p(y \mid x, q)$ directly, conditioning on a structured query $q$ alongside the history $x$ via a single deterministic forward pass.

inference time without any task-specific training. Currently, the only models offering zero-shot capabilities for EHR data are autoregressive (AR), generative models, which rival supervised approaches across diverse outcomes (Waxler et al., 2025; Renc et al., 2024; Rajamohan et al., 2025; Shmatko et al., 2025; Renc et al., 2025; McDermott, 2025). Much like large language models (LLMs), AR EHR models are trained to predict the next medical observation in a patient's longitudinal record. Unlike LLMs, which answer novel questions directly through task prompting, AR EHR models enable zero-shot inference without prompting through simulation: given a patient's history, they generate many possible "synthetic futures" and estimate the probability of an outcome by aggregating over these trajectories (Figure 1a).

While these models achieve zero-shot inference, they have three critical problems. First, simulation is extremely expensive: each prediction requires simulating many trajectories, so inference alone over even small datasets can take thousands of GPU hours (Solo et al., 2026). Second, the estimates are statistically noisy and struggle on rare outcomes: AR models count occurrences across $K$ sampled trajectories, so for a rare event with probability $p \ll 1$ the expected count $Kp$ may be tiny even for large $K$, yielding high variance and low sensitivity (Waxler et al., 2025; Renc et al., 2025). Third, AR models are not promptable: as their only input is the patient's history, they structurally *cannot* produce task-sensitive internal representations.

Here, we introduce **EveryQuery**, a promptable EHR foundation model that enables zero-shot prediction without simulation by directly predicting the outcome for a patient's medical data and task prompt in a single forward call (Figure 1b). It is both faster and more accurate than AR models, especially on low-prevalence tasks. EveryQuery is trained via *task-conditioned pretraining*: randomly constructed (task, patient context) pairs are sampled dynamically during pretraining, and the model is trained to map them directly to the correct outputs, which are computed automatically from the complete records. Because the pretraining task distribution has full support over all tasks expressible at inference time, the model learns to map any task/patient combination to its target output in a zero-shot manner. We validate on MIMIC-IV (Johnson et al.), NWICU (Moukheiber et al., 2024), and a large academic medical center ("AMC"), and demonstrate:

1. **Zero-shot performance.** EveryQuery outperforms an AR baseline across hundreds of randomly sampled tasks, with win rates of 84.6% on MIMIC-IV, 58.0% on AMC, and 61.3% on NWICU (Wilcoxon signed-rank $p < 10^{-4}$ across datasets). While AR models are error-prone on rare events, EveryQuery is prevalence-invariant.

2. **Efficient inference.** For a single prediction task, EveryQuery is approximately *5,500 to 10,000 times faster* than the AR baseline.

3. **Prompt specificity.** EveryQuery produces representations richly structured across all axes of the input task and patient, showing that task prompts directly inform the learned geometry.

## 2. The EveryQuery Method

**Task-conditioned pretraining.** Driven by the insight that clinical prediction tasks can be expressed in a simple, structured form, we sidestep trajectory sampling entirely. EveryQuery learns representations of the patient history conditioned on the task by taking both $\mathbf{x}$ and $q$ as input and predicting the answer directly, $p_\theta(y \mid \mathbf{x}, q)$ (Figure 1b). To answer *any* expressible downstream task zero-shot, we pretrain over (patient input, query) pairs where queries are randomly sampled from a full-support distribution. This scheme is fully task-agnostic, yielding an identical "API" to AR models: the model is pretrained over unlabeled EHR sequences and then answers arbitrary expressible tasks at inference. EveryQuery relates to instruction tuning (Sanh et al., 2021; Wei et al., 2021) and multitask EHR pretraining (Steinberg et al., 2023; Bertsimas & Ma, 2024; Steinberg et al., 2021), but unifies all tasks into one structured input format with a direct prediction objective and specifies tasks as inputs rather than fixed output heads (Appendix A).

**Query definition.** A patient's tokenized record is $\mathbf{x} = (x_1, \ldots, x_L)$ with $x_i \in \mathcal{V}$, where $\mathcal{V}$ is a unified vocabulary of medical codes following the MEDS standard (Arnrich et al., 2024) and elapsed time $t_j$ is recovered by accumulating time-delta tokens. We formalize a *query* as $q = (c, \Delta t)$, where $c \in \mathcal{V}$ is a target code and $\Delta t$ a prediction horizon in days. The binary answer label is $y(q, \mathbf{x}) = \mathbb{1}[\exists\, j > L : x_j = c \text{ and } t_L < t_j \leq t_L + \Delta t]$. When the record ends before $t_L + \Delta t$, the outcome is censored, which EveryQuery handles via a dedicated head. This captures the majority of clinical prediction tasks, which reduce to "will code $c$ occur within the next $\Delta t$ days?"; richer query languages are a natural extension (§5).

**Architecture.** EveryQuery uses a single bidirectional transformer backbone, ModernBERT-base[1], trained from scratch. Bidirectional attention lets the query token attend to and from the full patient context, allowing the representation to be task-conditioned. The query $q = (c, \Delta t)$ is encoded as two tokens: the code $c$ maps to an embedding $\mathbf{e}_c \in \mathbb{R}^d$ via the shared code embedding layer, and the duration $\Delta t$ passes through a small MLP to produce $\mathbf{e}_{\Delta t} \in \mathbb{R}^d$, an encoding continuous across horizons. The two query tokens are prepended to the history to form $[\mathbf{e}_c, \mathbf{e}_{\Delta t}, x_1, \ldots, x_L]$, processed jointly so query and patient tokens attend to one another. The final-layer hidden state at the code-token position is the task-conditioned representation; two MLPs map it to $\hat{y}_{\text{occurs}} = P(\text{event} \mid \mathbf{x}, q)$ and $\hat{y}_{\text{cens}} = P(\text{censored} \mid \mathbf{x}, q)$.

**Training and inference.** Each training example pairs a patient history with an independently sampled query. Only prediction times with a minimum number of preceding events are eligible. Codes are sampled uniformly from a subset of $\mathcal{V}$, and durations from a log-uniform distribution from 1 day to 5 years. A query is censored ($y_{\text{cens}} = 1$) if the record contains fewer than $\Delta t$ days of data after the prediction time; otherwise the occurrence label is computed from the actual future record. EveryQuery is optimized with $\mathcal{L} = \mathcal{L}_{\text{cens}} + \lambda \mathcal{L}_{\text{occurs}}$, where $\mathcal{L}_{\text{cens}}$ is a BCE over all samples and $\mathcal{L}_{\text{occurs}} = (1 - y_{\text{cens}}) \cdot \text{BCE}(\hat{y}_{\text{occurs}}, y_{\text{occurs}})$ is masked by censoring, so censored samples contribute only to the censoring loss. At test time the user specifies $q = (c, \Delta t)$ and EveryQuery produces $\hat{y}_{\text{occurs}}$ in a single deterministic forward pass, with no aggregation across trajectories. Full hyperparameters are in Appendix C.

## 3. Experiments and Results

**Setup.** We train and evaluate on three EHR datasets, each preprocessed into the MEDS format (Arnrich et al., 2024; McDermott et al., 2024): MIMIC-IV (v2.2) (Johnson et al.),

---

[1] https://huggingface.co/answerdotai/ModernBERT-base

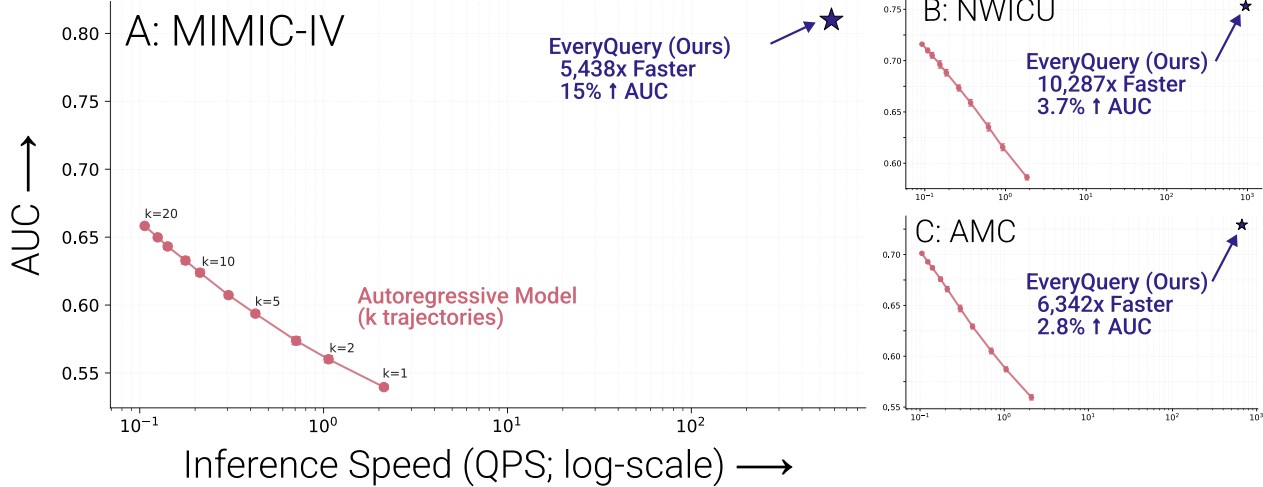

*Figure 2.* **Pareto frontier of zero-shot AUC vs. inference speed.** EveryQuery offers significant improvements in zero-shot task performance (macro-averaged AUROC, $y$-axis, higher is better) and inference throughput (queries per second, $x$-axis, higher is better) versus autoregressive, simulation-based inference regardless of the number of trajectories used, across *(A)* MIMIC-IV, *(B)* NWICU, and *(C)* AMC. Reported speed and AUC deltas are at the $k = 20$ setting. Error bars around AR numbers indicate variance in macro-averaged AUROC due to trajectory sampling, not variance across individual tasks, which is much larger.

NWICU (Moukheiber et al., 2024), and a private dataset from a large academic medical center (AMC). They vary substantially in population, care setting, and scale (240K, 24K, and 1.9M patients after preprocessing). We compare against MEDS-EIC-AR (McDermott, 2025)[2], an open-source "everything-is-code" AR model that fuses each clinical code with its decile-binned value into a single token (Lee et al.; Guo et al.); it generates $K = 20$ trajectories per patient, and both models operate on the same effective 256-token context at inference. For each dataset we draw 100 random codes and pair each with eight horizons (1, 3, 6 months; 1–5 years), except NWICU which lacks data beyond 2 years. After filtering tasks without both positive and negative labels, this yields hundreds of evaluation tasks spanning labs, diagnoses, medications, infusions, fluid outputs, and procedures. We report AUROC and emphasize aggregate per-dataset comparisons (win rates, signed-rank tests), as many tasks are rare events (Appendix B).

### 3.1. Zero-shot performance, especially on rare events

EveryQuery wins on a majority of tasks across all three datasets: 84.6% on MIMIC-IV, 61.3% on NWICU, and 58.0% on AMC, all significantly above chance (Wilcoxon signed-rank $p < 10^{-4}$). It achieves mean AUCs of 81.0%, 75.3%, and 72.9% respectively. As shown in Figure 2, EveryQuery also improves raw macro-averaged per-task AUROC: mean $\Delta$AUC reaches $+15.4\%$ (95% CI $[14.1, 16.7]$) on MIMIC-IV, with smaller but positive margins on NWICU ($+3.7\%$ $[2.23, 5.23]$) and AMC ($+2.3\%$ $[0.9, 3.6]$), consis-

tent with the tighter win margins on those datasets.

**Prevalence-invariant prediction.** EveryQuery's advantage is greater for rare events (Appendix E, Figure 4). On all three datasets, $\Delta$AUC is negatively correlated with prevalence (Spearman $\rho = -0.35$ on MIMIC-IV, $-0.51$ on NWICU, $-0.16$ on AMC; all $p < 10^{-4}$). Decomposing this, the AR model's AUC is positively correlated with prevalence everywhere ($\rho = +0.60, +0.44, +0.50$; all $p < 10^{-4}$), showing degraded discrimination on rare events, while EveryQuery's AUC shows little or no dependence ($\rho = +0.01, -0.13, +0.17$). The rare-event advantage thus reflects EveryQuery's prevalence-invariant performance rather than improved performance on rare events specifically.

### 3.2. Thousands of times more efficient

EveryQuery requires a single deterministic forward pass per (patient, query) pair, whereas the AR baseline generates $K = 20$ full trajectories, each with hundreds of events, per patient. EveryQuery is therefore dramatically faster on all datasets, with speedups of approximately 5,500 to 10,300× in queries-per-second at inference (Figure 2). Because AR generation is a one-time cost amortized across tasks while EveryQuery's cost scales linearly, the effective speedup shrinks as more tasks are evaluated simultaneously; given the magnitude of the speedup, however, it would take thousands of simultaneous predictions to favor AR, far beyond typical clinical workloads. Variance-reduced AR estimators (Solo et al., 2026) narrow this gap by roughly an order of magnitude but remain hundreds of times slower than direct prediction.

---

[2]https://github.com/mmcdermott/MEDS_EIC_AR

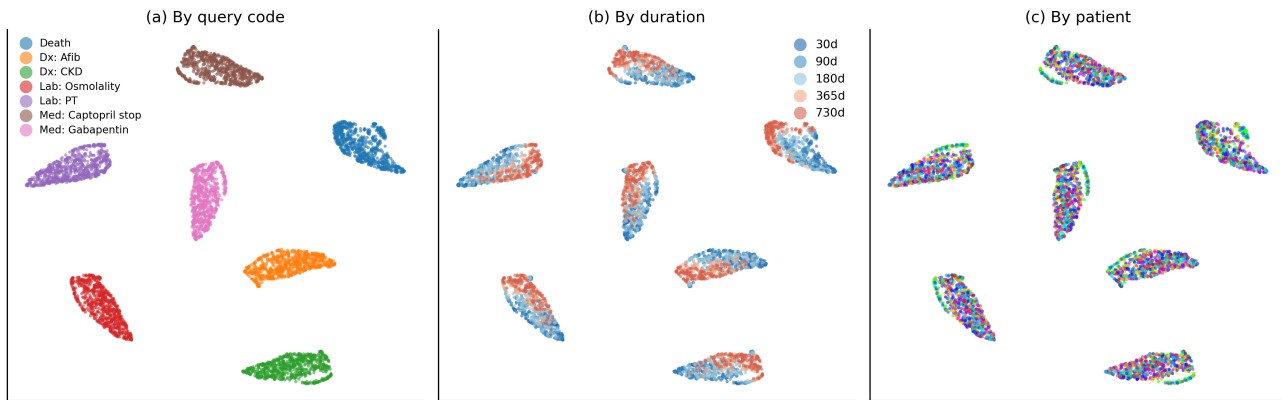

*Figure 3.* **EveryQuery representations are organized by task.** UMAP projection of MIMIC-IV embeddings, colored by query code (left), duration horizon (middle), and patient (right). Code identity produces distinct clusters; horizon is sequentially ordered within each cluster; patient identity produces less visible clustering but retains deep structure (§3.3).

### 3.3. Task-promptable representations

We ask whether EveryQuery genuinely conditions its representations on the query, or learns a fixed patient embedding with a task-specific readout. We extract the final-layer hidden state at the query-token position for all (patient, query) pairs on MIMIC-IV and analyze the geometry. Figure 3 shows a UMAP projection colored by query code, horizon, and patient. Code identity produces distinct, cleanly separated clusters; within each cluster, embeddings are sequentially ordered by horizon, suggesting a chronological representation of time. Patients also cluster reliably within *and across* queries: within a single code cluster the relative cosine-similarity matrix of aggregate patient embeddings is near-perfectly correlated ($\rho > 0.9$) across other codes. While the prompt code and duration dominate the macro structure, patients are consistently structured within code clusters, demonstrating rich macro and micro structure. These patterns hold across all datasets.

## 4. Probing the Limits of Task Conditioning

We stress-test EveryQuery along four axes; full details and tables are in Appendix D.

**Monotonicity in the horizon.** Predicted probabilities should be non-decreasing in $\Delta t$. Though not architecturally enforced, EveryQuery's predictions are monotonic across consecutive horizon pairs for 96.1% of (patient, code) trajectories on MIMIC-IV, 94.1% on AMC, and 92.8% on NWICU, degrading only modestly at the longest transitions.

**Composite queries.** We evaluate 30-day readmission on MIMIC-IV, a disjunction over 70 admission codes that EveryQuery cannot natively express. Querying all 70 codes and aggregating (Appendix D) reaches AUC $\approx 0.65$, at parity with the AR baseline (0.64), so EveryQuery extends to out-of-language tasks without catastrophic failure.

**Robustness to baseline choice.** We additionally compare against ETHOS (Renc et al., 2024), an independently published AR model tailored to MIMIC-IV whose smaller, coarser vocabulary and longer context favor it. On a 40-task suite mapped to the closest ETHOS-expressible task (Appendix D), EveryQuery wins 23, ties 3, and loses 14 (57.5%), remaining competitive even against this MIMIC-specific model while thousands of times faster.

**Adversarial queries.** Withholding 100 codes from query sampling during pretraining (they still appear in histories) probes generalization from patient context to task query. EveryQuery still beats AR on MIMIC-IV (74.7% vs. 84.6% standard) and AMC (54.5% vs. 58.0%), but drops to 34.4% on NWICU, the smallest corpus, exposing query-space overfitting as a risk on small datasets.

## 5. Discussion and Conclusion

Our evaluations demonstrate that task-conditioned pretraining is an advantageous alternative to autoregressive generation for EHR foundation models, winning on a majority of zero-shot tasks across three datasets while running more than $5,000\times$ faster and yielding a prompt-specific embedding space.

EveryQuery has two main limitations that motivate future work. First, while expressive, its query language is not universal: composite tasks spanning many codes (e.g., readmission, §4), event-bounded durations, complex future conditionals, and non-binary outcomes lie outside its current scope; extending the query language, e.g., using a full ACES (Xu et al., 2024) configuration as a query during pretraining, is a key direction. Second, EveryQuery's uniform sampling over codes, durations, and contexts may not represent the clinically meaningful subset of tasks. We see task-conditioned pretraining as a compelling direction for EHR foundation models and beyond.

## Acknowledgments

MBAM gratefully acknowledges support from a Berkowitz Postdoctoral Fellowship, of the Ivan and Francesca Berkowitz Family Living Laboratory Collaboration at Harvard Medical School and Clalit Research Institute.

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

## A. Comparison of EHR Foundation Models

EveryQuery draws on multitask pretraining, a strong representation-learning paradigm for EHR data (Steinberg et al., 2023; Bertsimas & Ma, 2024; Steinberg et al., 2021). MOTOR (Steinberg et al., 2023) pretrains thousands of time-to-event predictions in parallel; M3H (Bertsimas & Ma, 2024) unifies multimodal multitask prediction; CLMBR (Steinberg et al., 2021) learns shared representations adapted via downstream heads. EveryQuery shares the premise that supervising on many tasks at once produces transferable representations, but realizes it through a different objective. Multitask models attach a separate output head per task and predict them simultaneously, which can induce negative transfer when gradients interfere (Standley et al., 2020). EveryQuery instead passes the task to the model as input and trains a single output to answer it, so tasks share an input-output format rather than competing in the output space. Because tasks are specified at inference rather than fixed at training, EveryQuery generalizes zero-shot to queries never seen during pretraining, whereas multitask models are limited to predefined heads.

**Additional related work.** Beyond AR EHR foundation models (Renc et al., 2024; 2025; Waxler et al., 2025; Pang et al., 2025), a longstanding body of work pretrains bidirectional or encoder-decoder representations on EHR data (Rasmy et al., 2021; Pang et al., 2021; Wornow et al., 2023; Yang et al., 2023; Steinberg et al., 2021) and adapts via finetuning, linear probing, or task-specific heads, including multitask models such as MOTOR (Steinberg et al., 2023) and M3H (Bertsimas & Ma, 2024). These are not capable of zero-shot inference and so are not appropriate comparisons here. Outside health, instruction tuning over many NLP tasks induces zero-shot transfer to unseen tasks (Brown et al., 2020; Sanh et al., 2021; Wei et al., 2021), and prompt-based conditioning has been applied to EHR generation (Wang & Sun, 2022) and multi-domain time series (Gao et al., 2024), though not to discriminative clinical prediction.

## B. Experimental Setup Details

Each dataset is preprocessed into the MEDS format (Arnrich et al., 2024; McDermott et al., 2024) and partitioned into disjoint train/validation/test splits at the patient level (Table 1). The MEDS-EIC-AR baseline (McDermott, 2025) has 102M parameters and a max sequence length of 1024 tokens, of which at most 256 are reserved for the input patient context and the rest for autoregressive generation; both EveryQuery and MEDS-EIC-AR therefore operate on the same effective 256-token context at inference, though the AR baseline benefits from longer contexts during training. For each dataset we draw 100 random codes and pair each with eight horizons (1 month, 3 months, 6 months, 1–5 years), except NWICU which lacks data beyond 2 years. We filter tasks without both positive and negative labels. Many tasks are rare events (prevalence $< 0.5\%$) whose individual AUC estimates carry substantial uncertainty, so we emphasize aggregate comparisons throughout.

*Table 1.* **Dataset statistics per split after preprocessing.** Vocabulary size is reported at the dataset level.

| | Subjects | | | | |
|---|---|---|---|---|---|
| **Dataset** | **Train** | **Val** | **Test** | **Total Events** | **Vocab** |
| MIMIC-IV | 200,773 | 25,059 | 15,059 | 779,733,629 | 11,476 |
| AMC | 1,651,030 | 291,538 | 27,019 | 2,250,292,868 | 21,289 |
| NWICU | 20,723 | 2,590 | 772 | 54,785,364 | 2,159 |

## C. Architecture and Training Details

Table 2 summarizes the training and architecture hyperparameters used for EveryQuery.

## D. Probing the Limits: Additional Details

**Monotonicity.** Monotonicity holds nearly universally over short horizons and degrades modestly at the longest transitions, but residual inconsistency is small in absolute terms across all three datasets. This result resonates with the clear sequential ordering of duration in the embedding space (Figure 3).

**Composite readmission.** 30-day readmission on MIMIC-IV resolves into a disjunction over 70 distinct admission codes in the MIMIC schema. We evaluate two aggregation strategies over single-code queries: (i) max aggregation, $\hat{p}_{\text{readmit}} = \max_i \hat{p}_i$, and (ii) conditional independence, $\hat{p}_{\text{readmit}} = 1 - \prod_{i=1}^{70}(1 - \hat{p}_i)$. Both reach AUC $\approx 0.65$, at parity with

*Table 2.* **EveryQuery training and architecture hyperparameters.**

| Hyperparameter | Value |
|---|---|
| *Optimization* | |
| Optimizer | AdamW |
| Learning rate | $10^{-5}$ |
| Betas | $(0.9, 0.999)$ |
| Epsilon | $10^{-8}$ |
| Weight decay | 0.05 |
| LR schedule | Cosine with warmup |
| Warmup steps | 2,000 |
| Max training steps | 40,000 |
| Actual training steps | 24,859 |
| Wall-clock time | 385 min |
| Batch size | 160 |
| Precision | 16-bit mixed |
| *Architecture* | |
| Backbone | ModernBERT-base (from scratch) |
| Hidden dimension | 768 |
| Transformer layers | 22 |
| Attention heads | 12 |
| Backbone parameters | $\sim$149M |
| Max sequence length | 256 |
| MLP dropout | 0.1 |
| Answer head hidden dim | 128 |
| Answer head activation | ReLU |
| Answer head layers | 2 |
| *Data* | |
| Num. query codes (pretraining) | 10,000 |
| Total vocabulary size | 11,467 |
| Min. context events per prediction | 50 |
| Sequence sampling strategy | TO_END |
| Padding side | RIGHT |
| Batch mode | SM |
| *Infrastructure* | |
| GPU | $1\times$ NVIDIA L40S (48 GB) |

the AR baseline (0.64). The lack of a strong improvement emphasizes the importance of future work extending the query language to support compositional tasks natively.

**Robustness to baseline choice.** ETHOS (Renc et al., 2024) uses a smaller, dataset-specific vocabulary representing events at a coarser level (e.g., drug class rather than specific drug) and a longer sequence length, both of which make its inference task easier and give it more contextual information than either our AR baseline or EveryQuery. Its fundamentally different vocabulary creates a mapping problem, so we manually constructed a graded mapping to the closest ETHOS-expressible task: 12 tasks admit an exact match and 28 require relaxation (parent-level diagnosis, drug class, approximate lab binning, or clinical proxy). On this 40-task suite, EveryQuery wins 23, ties 3, and loses 14 (57.5%).

**Adversarial queries.** EveryQuery randomly samples *any* task expressible over the full vocabulary during pretraining; there is no realistic deployment setting where a clinically relevant code would be withheld from query sampling, since the vocabulary is fixed. Nonetheless, withholding 100 random codes from query sampling probes whether EveryQuery learns a generalizable code representation spanning patient context and task query. Win rates drop to 74.7% (MIMIC-IV), 54.5% (AMC), and 34.4% (NWICU). The NWICU result, the only case of underperformance, indicates that query-space overfitting is a significant risk on smaller datasets and a key target for future work.

# E. Prevalence Analysis

Figure 4 shows the per-task advantage of EveryQuery over the autoregressive baseline as a function of event prevalence on all three datasets. On every dataset, $\Delta$AUC is negatively correlated with prevalence, reflecting EveryQuery's prevalence-

invariant accuracy versus the AR baseline's degradation on rare events (§3.1).

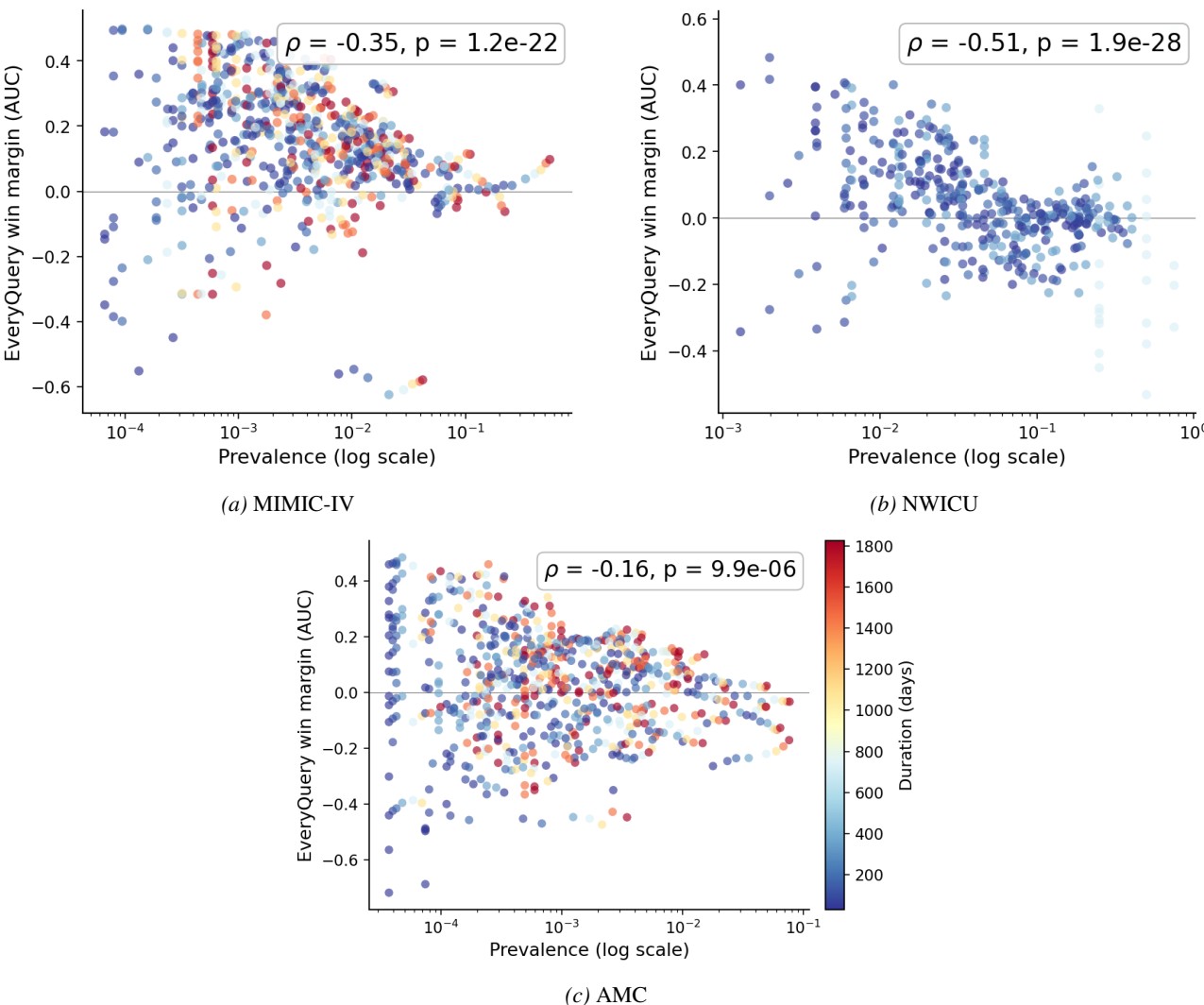

*(a)* MIMIC-IV

*(b)* NWICU

*(c)* AMC

*Figure 4.* **EveryQuery advantage vs. event prevalence across three datasets.** Each point is one (code, horizon) task; the $y$-axis shows EveryQuery AUC minus AR AUC, the $x$-axis shows prevalence (log scale), and color encodes the prediction horizon. On all three datasets $\Delta$AUC is negatively correlated with prevalence, reflecting EveryQuery's prevalence-invariant accuracy versus AR degradation on rare events.

# F. Full Results by Task

Per-task AUROC gaps ($\Delta$ = EveryQuery − AR) across follow-up horizons for all evaluation tasks, sorted by prevalence, are available in the accompanying preprint. Many tasks have low prevalence; individual AUC estimates for rare events should be interpreted with appropriate uncertainty.

