# OpenReview forum: "EveryQuery: Zero-Shot Clinical Prediction via Task-Conditioned Pretraining"
_ICML.cc/2026/Workshop/FMSD — FMSD @ ICML 2026 Poster_

### Official Review · Reviewer_aXMu · 2026-05-15
**A simple and promising  approach to zero-shot EHR foundation modeling**

**Rating:** 7
**Confidence:** 4

**Review:**

## Summary

This paper introduces EveryQuery, a task-conditioned foundation model for zero-shot clinical prediction over structured EHR data. The core idea is to represent a clinical prediction task as a structured query combining target medical code and prediction horizon, and to train a bidirectional transformer to directly estimate whether the queried event occurs within that horizon. This contrasts with autoregressive EHR foundation models, which generate synthetic future trajectories and estimate event probabilities by counting occurrences across sampled futures.

The paper evaluates EveryQuery on MIMIC-IV using 200 randomly sampled `(code, duration)` prediction tasks and compares against an autoregressive “everything-is-code” baseline. The authors report that EveryQuery outperforms the autoregressive baseline on 85% of tasks, achieves a substantially higher mean AUROC, generalizes to held-out query codes and durations, and is much faster at inference. The paper also argues that EveryQuery is more robust for rare events because it avoids the sampling noise and probability quantization inherent in finite trajectory-based inference.

Overall, I find the central approach compelling. Task-conditioned pretraining is a natural and elegant way to make structured EHR models more queryable, efficient, and broadly reusable. However, I found several aspects of the empirical evaluation difficult to interpret. In particular, the rare-event and “prevalence-invariance” claims rely heavily on AUROC and correlations with prevalence, while PR-based metrics, calibration, task-level uncertainty, and additional ablations would be needed to fully support the strongest claims.

## Strengths

The main strength of the paper is the formulation and the simplicity relative to autoregressive models. It is simple, useful, and well aligned with the workshop’s focus. The proposed model also directly addresses the limitation of autoregressive EHR models: inference via future trajectory sampling can be expensive and statistically noisy, especially for rare outcomes.

The method is also appealing from a practical standpoint. A single forward pass is much more convenient than sampling many synthetic futures, especially if a user wants to ask a specific clinical question at inference time. The reported efficiency gains are therefore meaningful, even if the exact magnitude depends on the number of concurrent tasks.

The paper is clearly written overall. The authors do a good job motivating why autoregressive generation is not always the right interface for clinical prediction, and the query-conditioned design is easy to understand. I also like the held-out code and held-out duration experiments, as these directly test whether the model is learning a reusable query-conditioned predictor rather than simply memorizing things.

The embedding and monotonicity analyses are useful sanity checks. They suggest that the model representation is genuinely modulated by the query and that the duration embedding has learned a mostly coherent temporal structure. I do not think these analyses fully prove the mechanism, but they are helpful supporting evidence.

## Areas for Improvement

My main concern is that the empirical claims are stronger than what is currently demonstrated. The headline result is promising, but the results are hard to interpret without more information about the sampled tasks. Since many randomly sampled EHR codes are likely to be rare, task-level AUROC estimates may have high uncertainty. The paper would be stronger with a table or appendix reporting, for each task or task group, the number of positives, prevalence, AUROC confidence intervals, AUPRC, and calibration metrics.

The “prevalence-invariance” claim should be reframed or supported with additional evidence. The paper argues that EveryQuery is prevalence-invariant because its AUROC is less correlated with event prevalence than the autoregressive baseline’s AUROC. This is suggestive, but AUROC alone is not sufficient for rare-event clinical prediction. For rare outcomes, AUPRC, prevalence-normalized AUPRC, precision at fixed recall, recall at fixed precision, and PPV among the top-k highest-risk patients would be more clinically informative. Raw AUPRC itself is prevalence-dependent, so I would recommend reporting both AUPRC and normalized lift over the prevalence baseline. It would also be very interesting to better understand the mechanism behind this improvement. For example, are the autoregressive baselines doing simpler ranking because of sampling noise, while EveryQuery has learned better risk signals?

The censoring treatment is reasonable in principle, but the role of the censoring head is unclear. If the occurrence loss is masked for censored examples and inference uses only the occurrence head, then the censoring head appears to be an auxiliary objective. The paper should clarify whether the censoring prediction is used at inference time, whether it improves occurrence prediction, and whether it risks encouraging the model to learn healthcare-utilization or follow-up patterns rather than event risk. An ablation comparing the current censoring head against simply excluding censored examples, or training without the censoring auxiliary loss, would be useful.

The query sampling strategy also needs more justification. Uniformly sampling query codes is defensible because EHR vocabularies are heavy-tailed and frequency-proportional sampling would overemphasize common codes. However, this sampling scheme may create many near-all-negative tasks and may not reflect clinically meaningful query distributions due to extreme class imbalance. I would love to see ablations against frequency-aware sampling, log-frequency/temperature sampling, or positive-enriched query sampling. This would help determine whether the gains are due to the task-conditioned architecture or partly due to the chosen query distribution.

The baseline comparison could be stronger. The autoregressive baseline uses 20 sampled rollouts, which seems likely to underperform for rare events, while previous works like [1] used 100 rollouts with 8192-token context. The paper correctly notes that finite sampling is problematic for rare events, but the comparison would be more convincing if it reported AR performance and runtime for multiple values such as 20, 100, and perhaps a larger value where feasible. This would make the accuracy-efficiency tradeoff clearer and avoid the impression that the baseline is disadvantaged by a low sampling budget.

I also think the “zero-shot” and “OOD generalization” framing should be made more precise. Held-out query codes still appear in patient histories, so the model has learned embeddings and contextual usage for those codes even if they were never used as query targets. That is still a useful generalization experiment, but it is not the same as generalizing to completely unseen codes or new code systems. Similarly, held-out duration extrapolation is interesting, but the paper should clarify exactly which durations were seen during training and how far the extrapolation extends beyond the training range. Ideally, it would be interesting to look at generalization to a different hospital system with differences in coding.

## Detailed Comments

- The paper’s rare-event argument is plausible, but AUROC can obscure clinically important differences when positives are rare. Please report AUPRC, prevalence-normalized AUPRC or lift over random, precision at fixed recall, recall at fixed precision, and/or top-k PPV. This is especially important because one of the main claims is improved performance for low-prevalence events.

- The current evidence shows that EveryQuery’s AUROC is less correlated with prevalence than the AR baseline’s AUROC. That is not necessarily the same as prevalence-invariant prediction. I suggest softening this claim or supporting it with additional metrics and stratified analyses.

- Please include the number of positives, number of negatives, prevalence, AUROC CI, and AUPRC for each task or at least for bins of task prevalence. Since the 200 tasks are formed from 40 codes × 5 horizons, the tasks are not fully independent. Statistical tests should account for correlations across tasks sharing the same code and patient population.

- The paper introduces a separate censoring head, but it is not clear how this head is used during inference. If it is only an auxiliary training signal, an ablation would help. I would suggest comparing the current model with censoring head, a model without censoring head but with masked occurrence loss, a model trained only on uncensored examples, and potentially inverse-probability weighting or other censoring-aware alternatives.

- In EHR data, missing future observation is often not random. It may reflect death, discharge, transfer, care fragmentation, or lack of future encounters. The paper should discuss whether censoring is informative and whether the model may exploit observation-process signals.

- Uniform code sampling is reasonable for coverage, but it may distort the training distribution and produce many very rare targets. Please compare against alternative query sampling strategies, such as frequency-proportional, log-frequency/temperature-based, or positive-enriched sampling.

- The AR baseline uses 20 rollouts, which is likely insufficient for very rare events. Please show how AR performance changes with a larger number of rollouts and possibly bigger context lengths. This would better support the claim that EveryQuery’s advantage is not merely due to an under-sampled baseline.

- The 40 random codes span labs, diagnoses, medications, infusions, fluid outputs, and procedures. These are heterogeneous prediction targets. Some may reflect clinical outcomes, while others may reflect care processes or measurement decisions. Please report performance by code type and discuss whether all sampled codes are clinically meaningful prediction targets.

- The paper evaluates on MIMIC-IV only. For a foundation model claim, even a small external validation experiment or temporal split would strengthen the evidence. If external data is not feasible, the limitation should be stated more clearly. I am very curious to see if it can generalize to a system where different coding is used.

- The readmission experiment is useful because readmission requires a disjunction over many admission codes. The fact that EveryQuery needs post-hoc aggregation highlights that the current query language is limited to single-code targets. This is not a fatal flaw, but it should be presented as a central limitation rather than only future work.

- The monotonicity analysis is helpful, but if the model is intended to predict event occurrence within increasing horizons, non-decreasing probabilities are a natural constraint. The authors could discuss whether monotonicity should be enforced architecturally or through a training penalty.

- The paper notes that AR generation is a one-time cost amortized across tasks, whereas EveryQuery scales linearly with the number of queries. This is important. A plot of runtime versus number of queried tasks would make the efficiency tradeoff clearer.

- The model is promptable in the sense of accepting structured queries. However, it does not yet support richer clinical prompts, natural language task descriptions, code sets, conjunctions, exclusions, numeric predicates, or competing risks. The paper should avoid implying broader promptability than is currently demonstrated. Another future direction could be whether we can fine-tune a medical language model to do this.

## Justification

I am positive on this submission overall. The paper proposes a clean and useful approach to zero-shot clinical prediction over structured EHR data, and it is highly relevant to the workshop. I think the audience would appreciate this. The task-conditioned formulation is nice, and the efficiency and simplicity make this a promising alternative to autoregressive trajectory sampling, which may not be the most natural interface for actual zero-shot prediction.

I have some concerns about the strength and clarity of the empirical evidence as mentioned above. But the paper can be substantially improved, and I feel the idea has a lot of potential. I recommend Accept.

## References

[1] Waxler, Shane, et al. “Generative medical event models improve with scale.” arXiv preprint arXiv:2508.12104, 2025.

---

### Official Review · Reviewer_tKyq · 2026-05-16
**a task-conditioned EHR model for zero-shot prediction of whether a target medical code occurs within a specified future horizon**

**Rating:** 6
**Confidence:** 3

**Review:**

pros:

 - The main empirical result is substantial within the tested setup: on 200 MIMIC-IV code-duration tasks, EveryQuery reports mean AUROC 0.84 versus 0.67 for MEDS-EIC-AR, wins on 85% of tasks, and gives confidence intervals and paired testing. Aggregate reporting is appropriate because many individual tasks are rare and noisy.
 - Held-out query-code and held-out duration experiments are useful probes of the structured query space. They support some transfer beyond exact training targets, but they do not prove generalization to unseen medical concepts because held-out codes still appear in patient histories and their embeddings are trained through the shared code vocabulary.

cons:

 - strongest “zero-shot arbitrary clinical prediction” framing is too broad for the evidence. Training uses direct supervised labels for randomly sampled single-code, single-horizon queries, and most evaluation tasks follow the same query grammar; a defensible claim is zero-shot with respect to downstream finetuning for single-code time-window queries, with limited evidence of transfer to held-out query targets.
 - baseline comparison is the largest threat to the causal interpretation. EveryQuery uses a 149M-parameter bidirectional ModernBERT-style model, while MEDS-EIC-AR has 102M parameters and uses K = 20 sampled trajectories; no larger-K, compute-matched, variance-reduced, matched-backbone, or task-specific supervised baseline is shown. This leaves open whether the gap comes from query conditioning, capacity, architecture, objective, or a weak sampling setting.
 - Probability quality is under-evaluated relative to the model’s stated purpose. EveryQuery directly estimates event probabilities, yet the primary metric is AUROC, which measures ranking rather than calibrated risk. Brier score, expected calibration error, calibration plots, and rare-event PR-AUC would be more informative for clinical use.